# PHI-1, an Endogenous Inhibitor Protein for Protein Phosphatase-1 and a Pan-Cancer Marker, Regulates Raf-1 Proteostasis

**DOI:** 10.3390/biom13121741

**Published:** 2023-12-04

**Authors:** Jason A. Kirkbride, Garbo Young Nilsson, Jee In Kim, Kosuke Takeya, Yoshinori Tanaka, Hiroshi Tokumitsu, Futoshi Suizu, Masumi Eto

**Affiliations:** 1Department of Molecular Physiology and Biophysics, and Kimmel Cancer Center, Jefferson Medical College, Thomas Jefferson University, 1020 Locust Street, Philadelphia, PA 19107, USA; 2Department of Molecular Medicine, Keimyung University School of Medicine, Daegu 42601, Republic of Korea; 3Department of Veterinary Medicine, Faculty of Veterinary Medicine, Okayama University of Science, Imabari 794-8555, Ehime, Japany-tanaka@ous.ac.jp (Y.T.); 4Applied Cell Biology, Graduate School of Interdisciplinary Science & Engineering in Health Systems, Okayama University, Okayama 700-8530, Okayama, Japan; 5Oncology Pathology, Department of Pathology and Host-Defense, Faculty of Medicine, Kagawa University, Kita-gun 761-0793, Kagawa, Japan; suizu.futoshi@kagawa-u.ac.jp

**Keywords:** proteostasis, protein phosphatase, Raf-1, PP1, PHI-1, PPP1R14B

## Abstract

Raf-1, a multifunctional kinase, regulates various cellular processes, including cell proliferation, apoptosis, and migration, by phosphorylating MAPK/ERK kinase and interacting with specific kinases. Cellular Raf-1 activity is intricately regulated through pathways involving the binding of regulatory proteins, direct phosphorylation, and the ubiquitin–proteasome axis. In this study, we demonstrate that PHI-1, an endogenous inhibitor of protein phosphatase-1 (PP1), plays a pivotal role in modulating Raf-1 proteostasis within cells. Knocking down endogenous PHI-1 in HEK293 cells using siRNA resulted in increased cell proliferation and reduced apoptosis. This heightened cell proliferation was accompanied by a 15-fold increase in ERK1/2 phosphorylation. Importantly, the observed ERK1/2 hyperphosphorylation was attributable to an upregulation of Raf-1 expression, rather than an increase in Ras levels, Raf-1 Ser338 phosphorylation, or B-Raf levels. The elevated Raf-1 expression, stemming from PHI-1 knockdown, enhanced EGF-induced ERK1/2 phosphorylation through MEK. Moreover, PHI-1 knockdown significantly contributed to Raf-1 protein stability without affecting Raf-1 mRNA levels. Conversely, ectopic PHI-1 expression suppressed Raf-1 protein levels in a manner that correlated with PHI-1’s inhibitory potency. Inhibiting PP1 to mimic PHI-1’s function using tautomycin led to a reduction in Raf-1 expression. In summary, our findings highlight that the PHI-1-PP1 signaling axis selectively governs Raf-1 proteostasis and cell survival signals.

## 1. Introduction

Normal embryonic development necessitates precise coordination of growth factor-induced processes such as cell proliferation, differentiation, apoptosis, and migration. Raf-1 (C-Raf) is a member of the mammalian Raf family and plays a specific role in phosphorylating MEK, ultimately leading to ERK1/2 phosphorylation [1]. Extensive research has elucidated the post-translational mechanisms governing the activation and inhibition of Raf-1 kinase [1,2,3,4]. Furthermore, Raf-1 also forms heterodimers with other kinases like ASK1, MST, and ROCK2, acting as an endogenous inhibitor for these kinases [4,5,6,7]. This inhibition, independent of Raf-1 kinase activity, plays a crucial role in the regulation of differentiation, apoptosis, and migration [5,6,7,8]. The significance of Raf-1 becomes evident in Raf-1 knockout mice, where enhanced apoptosis in developing tissues leads to embryonic death [9,10]. The Raf-1 gene promoter shares elements with housekeeping genes [11]. Cellular Raf-1 protein stability is regulated through ubiquitin–proteasome pathways [12,13,14]. Disturbances in Raf-1 expression levels can disrupt organogenesis during early development [15]. Raf-1 upregulation is essential for erythroid development [16] and is also implicated in certain cancers, including leukemia, squamous cell carcinoma, ovarian cancer, and melanoma [17,18,19,20]. Thus, Raf-1 proteostasis intricately links physiological regulation and pathological events in diverse cell types.

Type-1 Ser/Thr phosphatase (PP1) is abundantly expressed and plays multifaceted roles in cellular processes. Cellular PP1 exists as holoenzymes consisting of a catalytic subunit with over 100 regulatory subunits, each conferring specific functions [21,22,23,24]. Increasing evidence indicates that endogenous PP1 inhibitor proteins target specific PP1 holoenzymes, thereby regulating distinct cellular events [23,25,26]. Many PP1 inhibitors are phospho-proteins, enabling them to mediate kinase signals to PP1 holoenzymes [26,27]. One such inhibitor is PHI-1 (phosphatase holoenzyme inhibitor-1), a transcript of the PPP1R14B gene, originally identified as a phospholipase C-neighboring gene [28,29]. PHI-1 is widely expressed in mammalian tissues [29] and is the most abundant PP1 regulator in HeLa cells [30]. Overexpression of PHI-1 has been associated with melanoma and ovarian cancer [31,32], and its expression levels are linked to prostate cancer and other malignancies [33,34,35,36,37,38]. However, the downstream signaling pathways modulated by PHI-1 remain largely unknown.

PHI-1 shares a conserved PP1 holoenzyme inhibitory (PHIN) domain with CPI-17, a myosin phosphatase inhibitor protein [25,39]. Similar to CPI-17, PHI-1’s inhibitory potency is enhanced when it is phosphorylated at Thr57 [29,40]. Phosphorylation of PHI-1 occurs in smooth muscle in response to G protein activation [41,42]. Recombinant PHI-1 can be phosphorylated by various kinases, including PKC, ILK, and ROCK [29,40], though the physiological kinases for PHI-1 are yet to be identified. Notably, PHI-1 inhibits PP1 holoenzymes such as myosin phosphatase and glycogen-bound PP1 [29,40,42]. Silencing PHI-1 expression in endothelial cells and HeLa cells suppresses cell migration by blocking cell edge retraction [43] or inhibits cell proliferation through downregulation of AKT [44] and/or dephosphorylation of stathmin1 [37]. However, PHI-1 does not appear to be involved in myosin phosphorylation, suggesting that myosin phosphatase is not a physiological target of PHI-1, at least in smooth muscle, endothelial, or HeLa cells [39,43,45].

Our study reveals the divergent role of PHI-1 in HEK293 cells as opposed to HeLa cells. HEK293 cells were specifically chosen due to their markedly lower PHI-1 expression levels compared to other cancer cells. The siRNA-induced knockdown of PHI-1 leads to enhanced cell proliferation and a reduction in apoptosis within HEK293 cells. These effects are accompanied by a concurrent increase in ERK1/2 phosphorylation and Raf-1 expression, which is attributed to the elevated protein stability. Our findings underscore the influence of a novel PP1-PHI-1 signaling axis on the proteostasis of Raf-1 in these cells.

## 2. Materials and Methods

Cell Culture and Gene Silencing: The human embryonic kidney cell line (HEK293) was obtained from ATCC (#CRL-1573) and cultured in Dulbecco’s modified eagle medium (DMEM; Mediatech, Manassas, VA, USA) supplemented with 5% fetal bovine serum (FBS, Mediatech, Manassas, VA, USA). Cells were used within passage 10. Gene silencing was carried out using two independent siRNA fragments for human PHI-1 (siGENOME ON-TARGET™ #D-026574-01) or (ON-TARGETplus SMARTpool™ #L-003601-00) obtained from Dharmacon. Both siRNA fragments yielded similar results, and data with siGENOME are presented in this article. siCONTROL (Dharmacon #D-001210-01, Lafayette, CO, USA) served as a non-targeting control. Knockdown experiments were performed with 50 pmol of siRNA and 2.5 µL of Lipofectamine 2000, following the manufacturer’s protocol. Cells were treated with the siRNA mixture in the presence of the growth medium for 72 h before conducting assays. For cells expressing enhanced yellow fluorescence protein Venus (Vns) [46] or Vns-tagged Raf-1 (Vns-Raf-1), siRNA treatment was carried out for 48 h before transfection. After replacing the medium with a fresh culture medium, a mixture of 0.5 µg of the ectopic DNA vector and FugeneHD transfection reagent (Roche, Indianapolis, IN, USA) was added to the cells. Transfection continued for an additional 24 h before conducting assays. Fluorometric caspase-3 assay was conducted using Fluorometric Caspase-3 Assay Kit DEVD-AMC (Sigma-Aldrich #CASP3F, St. Louis, MO, USA) following the manufacturer’s protocol. The fluorescence intensity released by the caspase-3 in the cell lysates was determined using Tecan Safire2.

Expression and Phosphorylation of Proteins: Protein expression and phosphorylation were assessed through immunoblotting followed by densitometry analysis. After treatment, cells were rinsed once with DPBS and fixed with 10% trichloroacetic acid (TCA). The cell debris was collected using a cell lifter, pelleted, and washed with diethyl ether twice. After drying, total proteins were extracted by homogenization with an SDS buffer containing 50 mM Tris-HCl (pH 8.0), 1% SDS, 1 mM EDTA, 1.2 mM sodium orthovanadate, and 4 mM Pefabloc™. The soluble fraction was collected and subjected to a BCA assay. Following the protein assay, the total protein solution was mixed with 2x Laemmli buffer (0.125 M Tris-HCl, pH 6.8, 4% SDS, 20% Glycerol, 10% 2-mercaptoethanol, 20% sucrose, and 0.04% bromophenol blue) and heated for 5 min at 95 °C. This protocol aimed to prevent interference with the protein assay by components in the Laemmli buffer. Twenty micrograms of total proteins were subjected to SDS-PAGE, followed by immunoblotting [47]. Primary antibodies used included anti-PHI-1 (1:5000, [29]), anti-smooth muscle α-actin (1:5000, Sigma-Aldrich #A5228), anti-P-ERK1/2 (1:10,000, Sigma-Aldrich #M8159), anti-total ERK2 (1:1000, Chemicon #AB3055, Temecula, CA, USA), anti-total Raf-1 (1:10,000, BD Transduction Lab #610151, Lexington, KY, USA), anti-phospho-Raf-1 (S338) (1:2000, Cell Signaling #9427, Danvers, MA, USA), anti-B-Raf (1:250, Merck #14-217, Darmstadt, Germany), anti-GFP (1:5000, Aves Lab #GFP1020, Davis, CA, USA), anti-HA (1:10,000, clone 12CA5), anti-CDK4 (1:2000, MyBioSource #MBS127817, San Diego, CA, USA), anti-PKN2 (1:2000, Cell Signaling #2612), anti-HSP70 (1:2000, Cell Signaling #4822), anti-P-MEK1/2 (1:2000, Cell Signaling #9121), and anti-GAPDH (1:5000, Novus #NB300-221, Centennial, CO, USA). Staining images were captured using the Alpha-innotech Fluorochem CCD Gel Imager, and band intensities were quantified using the attached AlphaEase FC software (http://genetictechnologiesinc.com/alpha/alpha_ease_fc.htm). GAPDH staining served as an internal loading control.

Reverse Transcription Quantitative Polymerase Chain Reaction (RT-qPCR): Total RNA was purified from cells after siRNA treatment using the QIAprep kit (Qiagen, Hilden, Germany). A 200 ng aliquot of total RNA was subjected to qRTPCR using the SyBR green-based Brilliant II QRTPCR kit (Stratagene, Santa Clara, CA, USA). The primer sets used were GCAATGAAGAGGCTGGTAGCTG/CGTGGTCAGCGTGCAAGCATTG for Raf-1 and AGTGGATCCTGGAGCAGCTCAC/GGTGTGCTCAGCTTCTGCATGC for PHI-1. The human histone H4 primer set was purchased from Realtimeprimers.com (Accessed on 7-November-2007). RT-qPCR was carried out using the Stratagene Mx3005P system with the following optimized conditions: reverse transcription for 30 min at 50 °C, followed by a 30-cycle, 3-step PCR with an annealing temperature at 55 °C. The cycle number at the threshold (Ct value) was obtained using MxPro software Ver. 4.10. The relative extent of mRNA was calculated by the ∆∆Ct method using the Ct value of histone H4 as an internal control [48].

Others: The cDNA insert of human Raf-1 was cloned from a HeLa cell cDNA library. The PCR fragment of human Raf-1 cDNA was cloned into a pRK5-derived pVns vector. The pVns vector was generously provided by Dr. Ian Macara, University of Virginia. The use of Vns cDNA was permitted by Dr. Atsushi Miyawaki, RIKEN Brain Science Institute, through a material transfer agreement. The DNA sequence of the insert was confirmed at the Cancer Genomics Facility at Thomas Jefferson University. Mean values ± SEM were obtained from at least three independent experiments, and the number of experiments (n) is indicated in each figure legend. A *p*-value below 0.05, determined by an unpaired Student’s *t*-test or ANOVA with Tukey’s test, was considered statistically significant. Statistical analysis of RT-qPCR data was performed using the method described by Yuan et al. [49].

## 3. Results

Expression of PHI-1 in cell culture was determined using RT-qPCR (Figure 1A). PHI-1 expression was relatively lower in HEK293 compared to two tumor-origin cell lines, HeLa and Panc1. To explore the non-cancerous roles of PHI-1, gene silencing assays were performed in HEK293 cells. To minimize off-target effects, SMART pool siRNA, a mixture of four siRNA fragments, was used for PHI-1 knockdown. The optimized knockdown condition consistently led to an approximately 80% reduction in PHI-1 protein (Figure 1B). Unlike the findings reported by Tountas et al. [43], which showed a bi-spindle phenotype in HeLa cells, siRNA knockdown of endogenous PHI-1 did not cause noticeable morphological changes in HEK293 cells. Instead, the proliferation of cells treated with PHI-1 siRNA significantly increased at 2 days and 3 days after transfection (Figure 1C, closed circle) compared to the control (open circle, Ctl). In parallel, caspase-3 activity was suppressed in cells transfected with PHI-1 siRNA at 2 days after knockdown (Figure 1D). These data suggest that PHI-1 in HEK293 cells mediates anti-proliferative signaling.

The densitometric analysis of immunoblotting, shown in Figure 2, revealed that a 72 h PHI-1 knockdown resulted in a 21-fold increase in ERK1/2 phosphorylation in HEK293 cells under the growth condition, without a significant increase in ERK1/2 expression (Figure 2). The increase in ERK1/2 phosphorylation was accompanied by a 7.2-fold elevation of Raf-1 expression (Figure 2). Phospho-Ser338-Raf-1, an indicator of active Raf-1, was partially increased compared to the total Raf-1 level. In sharp contrast, PHI-1 knockdown had no effect on the expression of other kinases, such as B-Raf, CDK4, or PKN2 (Figure 2). Thus, endogenous PHI-1 selectively regulates Raf-1 expression among kinases.

Figure 3 shows EGF-induced ERK1/2 phosphorylation. HEK293 cells were treated with PHI-1 siRNA and then harvested in a medium with 1% serum before a 10 min stimulation with EGF. PHI-1 knockdown amplified ERK1/2 phosphorylation induced by 10 ng/mL EGF (Figure 3A,B). ERK1/2 phosphorylation was associated with MEK1/2 phosphorylation and Raf-1 upregulation (Figure 3A,B). Pretreatment with U0126 eliminated the EGF-induced ERK1/2 phosphorylation in both the control and PHI-1-deprived cells (Figure 3C), indicating that ERK1/2 hyperphosphorylation upon PHI-1 knockdown is due to MEK1/2 activation. These results suggest that endogenous PHI-1 functions as a negative regulator for ERK1/2 phosphorylation, and depletion of PHI-1 enhances the level of an active Raf-1, causing ERK1/2 hyperphosphorylation.

We investigated whether PHI-1 downregulation impacts Raf-1 protein stability using the ectopic vector for Vns-tagged Raf-1 (Vns-Raf-1), under the control of the CMV promoter (Figure 4A). The results revealed that PHI-1 knockdown led to a notable increase in both the endogenous Raf-1 kinase and the ectopic Vns-Raf-1 fusion protein (Figure 4A, top), with an approximately 1.5-fold elevation. Notably, the expression of Vns protein lacking the Raf-1 insert remained unaltered (Figure 4A, second from the top). It is worth mentioning that the activity of the CMV promoter, which governs Vns-Raf-1 transcription, remained insensitive to PHI-1 expression. Consistently to the PHI-1 knockdown data, the co-expression of wild-type PHI-1 resulted in a reduction of Vns-Raf-1 expression (Figure 4B). This suppressive effect of PHI-1 was abolished when the Thr57 phosphorylation site, which is necessary for the PP1 inhibition, was substituted with alanine (Figure 4B). Furthermore, a substantial 80% reduction in PHI-1 mRNA through siRNA caused a statistically significant yet subtle increase in Raf-1 mRNA levels (1.3 fold compared to the control, *p* < 0.05, n = 3). In addition, Ras activity is insensitive to PHI-1 knockdown (Appendix A). These data suggest that PHI-1 negatively regulates the Raf-1 protein level with minimal impact on transcription or Ras binding.

PHI-1 downregulation is expected to increase PP1 activity due to dis-inhibition. We examined the role of cellular PP1 activity in regulating Raf-1 proteostasis using tautomycin (TAUTO), a PP1 inhibitor compound [50] (Figure 5). To mimic the chronic inhibition of PP1 by PHI-1, HEK293 cells were treated overnight with a low dose of TAUTO (300 nM). As shown in Figure 5, the inhibition of endogenous PP1 by pre-treatment with TAUTO decreased the Raf-1 protein level in HEK293 cells without changing PHI-1 expression. These results suggest that the PP1-PHI-1 signaling axis is responsible for regulating Raf-1 proteostasis.

## 4. Discussion

Figure 6 presents a model illustrating the regulation of Raf-1 kinase expression by PHI-1. Depletion of cellular PHI-1, which activates endogenous PP1, interferes with the degradation of Raf-1 protein. In alignment with this, PP1 inhibition by TAUTO mirrors PHI-1’s function, resulting in Raf-1 downregulation. Consequently, the PP1 activity controlled by PHI-1 emerges as a determinant in Raf-1’s proteostasis within HEK293 cells. Notably, the effect of TAUTO is less pronounced than PHI-1 knockdown. This disparity might stem from the broad inhibition of multiple PP1 holoenzymes by TAUTO, leading to off-target effects on Raf-1 expression. Conversely, PHI-1 likely interacts with a select subset of PP1 holoenzymes, consisting of a PP1 catalytic subunit and the regulatory subunit (s). Previously, we reported that MYPT1, a myosin-targeting subunit of PP1, directly interacts with Raf-1 [51]. However, it seems that Raf-1 proteostasis is independent of the MYPT1 interaction, as PHI-1 had no influence on the myosin phosphatase holoenzymes [39,45].

Multiple lines of evidence suggest that Raf-1 expression is regulated through ubiquitination and subsequent degradation via the proteasome pathway [12,13,14]. Notably, HSP90 inhibition with geldanamycin treatment leads to Raf-1 misfolding, followed by degradation [52]. This points to the existence of a quality control system that oversees Raf-1 kinase maturation (illustrated in Figure 6). Mature Raf-1, in particular, undergoes Ser621 autophosphorylation, protecting it from degradation [12]. PHI-1’s role likely does not extend to the regulation of Ser621 dephosphorylation since inhibiting the Ser621 phosphatase would be expected to increase Raf-1 expression. Intriguingly, PHI-1 downregulation does not lead to the misfolding of Raf-1 kinase. Instead, it enhances the activity of the kinase, as evidenced by increased phosphorylation of MEK1/2 and ERK1/2. As a result, the PHI-1-PP1 signaling pathway appears to dictate the turnover of active Raf-1 kinase. Raf-1 is known to be phosphorylated at various sites [1], many of which still await definitive characterization. It is possible that specific phosphorylation at known or yet-to-be-discovered sites is dephosphorylated by a PP1 holoenzyme. Consequently, PP1 inhibition by PHI-1 may elevate the phosphorylation, subsequently triggering kinase degradation. Thus, multiple phosphorylation sites may regulate positively and negatively Raf-1 stability regulating the proteostasis

It is essential to highlight that, in addition to its influence on MEK kinase activity, Raf-1 directly interacts with ASK1, MST, and ROCK2, regulating apoptosis and migration [4,5,6,7]. The regulation of these Raf-1-binding kinases is independent of Raf-1 kinase activity but relies on direct interaction. Consequently, fluctuations in Raf-1 kinase levels, induced by the PHI-1-mediated proteostasis pathways, potentially impact downstream kinase signaling, in addition to the ERK1/2 pathway.

PHI-1 is expressed ubiquitously, both in mature and embryonic tissues and mammalian cell cultures [29,53], and was identified as the most expressed PP1 regulator in HeLa cells [30]. Raf-1, similarly, is a ubiquitous protein. Its knockout leads to increased apoptosis in the embryonic liver and other tissues, ultimately resulting in embryonic death [9,10]. Although Raf-1 expression remains consistent during the early stages of inner ear organogenesis, deviations lead to abnormal development [15]. Nevertheless, Raf-1 expression levels vary across tissues and fluctuate under specific conditions. For instance, Raf-1 is downregulated in differentiating erythroid cells [16]. Furthermore, interleukin-2 stimulation enhances Raf-1 mRNA and protein levels in freshly isolated human T-cells [54]. Our findings suggest that the PP1-PHI-1 signaling pathway plays a pivotal role in maintaining Raf-1 expression levels within cells.

The initial report by Tountas et al. indicated that PHI-1 knockdown suppressed the migration and retraction of HeLa cells and endothelial cells without any notable effect on ERK1/2 phosphorylation [43]. However, recent research conducted over the past decade has continuously reinforced PHI-1’s significance in cell proliferation, consistently linking it to unfavorable clinical outcomes. Most notably, PHI-1 upregulation has been consistently observed across various malignancies, including ovarian clear cell carcinoma, chronic lymphocytic leukemia, prostate cancer, glioblastoma, triple-negative breast cancer, and uterine corpus endometrial carcinoma [32,33,34,37,38,55]. Furthermore, strong correlations between heightened PHI-1 expression levels and poorer prognoses have been established across various cancer types [33,35,37,38].

Of particular interest, PHI-1’s role appears to be dual in nature. While it promotes proliferation in cells with elevated PHI-1 levels, such as HeLa cells and other tumor cells [33,35,37,38], it simultaneously exerts a negative influence on cells with lower PHI-1 expression, such as HEK293. Notably, PHI-1 knockdown in HeLa cells suppressed proliferation, partly due to the attenuation of AKT signaling and/or stathmin phosphorylation [37,44]. One possible interpretation is that the positive effects of PHI-1 knockdown on HEK293 cell proliferation may be outweighed by the negative signaling in cancer cells with higher PHI-1 expression. This duality suggests that cells with high PHI-1 expression become reliant on it, akin to an addiction. In fact, the knockdown of PHI-1 in Panc1 cells, which express PHI-1 to a relatively greater extent compared to HEK293, resulted in a subtle elevation of Raf-1 expression (1.75 ± 0.28-fold vs. control, *p* < 0.05). The precise underlying mechanism governing PHI-1’s diverse cellular functions remains a subject of ongoing investigation, just as the critical endeavor of identifying the specific PP1 holoenzymes regulated by PHI-1 in various cellular contexts continues to be explored.

PHI-1 belongs to the CPI-17 family, which encompasses CPI-17, GBPI, and KEPI [25,26,39,56]. Additionally, the realm of endogenous inhibitor proteins for PP1 plus PP2A is extensive [26,57]. Each of these PP1 inhibitor proteins is phosphorylated and is believed to modulate a specific subset of PP1 holoenzymes, orchestrating kinase and phosphatase signaling pathways, albeit with limited information available. In this study, we have demonstrated the multifaceted roles of PHI-1 that are contingent on the specific cell types examined. Our findings underscore the complexity of protein phosphorylation signaling and emphasize the necessity of delving further into the intricate world of PP1 inhibitor proteins to gain a comprehensive understanding of these regulatory networks and pathological impacts induced by their dysregulation.

## 5. Conclusions

Our study, which characterizes a new role of PHI-1 in Raf-1 proteostasis, reveals the multifaceted functions of PHI-1 in cell signaling, demonstrating its cell type-dependent influence. This insight underscores the intricate contributions of cellular PP1 in regulating both normal cellular processes and pathological conditions.

## Figures and Tables

**Figure 1 biomolecules-13-01741-f001:**
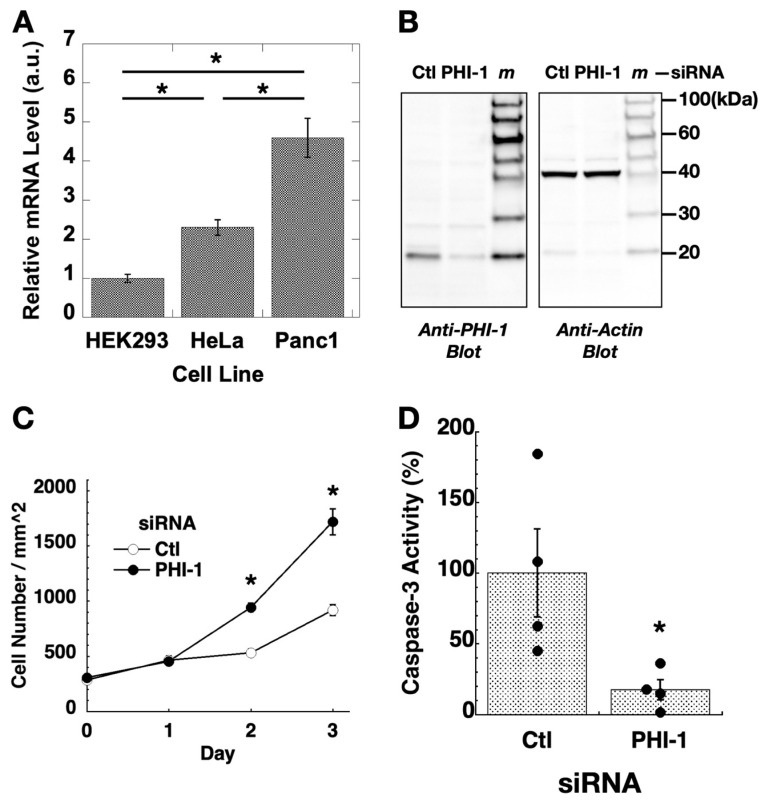
Expression and knockdown of PHI-1 in cell culture. (**A**) Relative levels of PHI-1 mRNA in cells were determined by RT-qPCR. Histone H4 was used as an internal control. Expression levels of PHI-1 mRNA are expressed as a relative value of the average of the PHI-1 mRNA expression level in HEK293 cells. Mean values ± SEM of relative mRNA levels calculated by ddCt method are shown. * indicates *p* < 0.05 by ANOVA with Tukey’s test (n = 3). (**B**) HEK293 cells were treated for 72 h with siRNA of control (Ctl) or PHI-1 and subjected to immunoblotting. *m* indicates the lane of the molecular marker. Original images can be found in Appendix A. (**C**) HEK293 cells were seeded on triplicate plates under the growth condition and transfected with siRNA on day 0. After 24 h, fresh media were given to the cells. Cell numbers in 15 view areas of each set were counted under a phase-contrast microscope. * indicates *p* < 0.05 vs. Ctl by *t*-test (n = 15). (**D**) HEK293 cells were transfected for 48 h and then subjected to caspase-3 assay. * indicates *p* < 0.05 vs. Ctl by *t*-test (n = 4).

**Figure 2 biomolecules-13-01741-f002:**
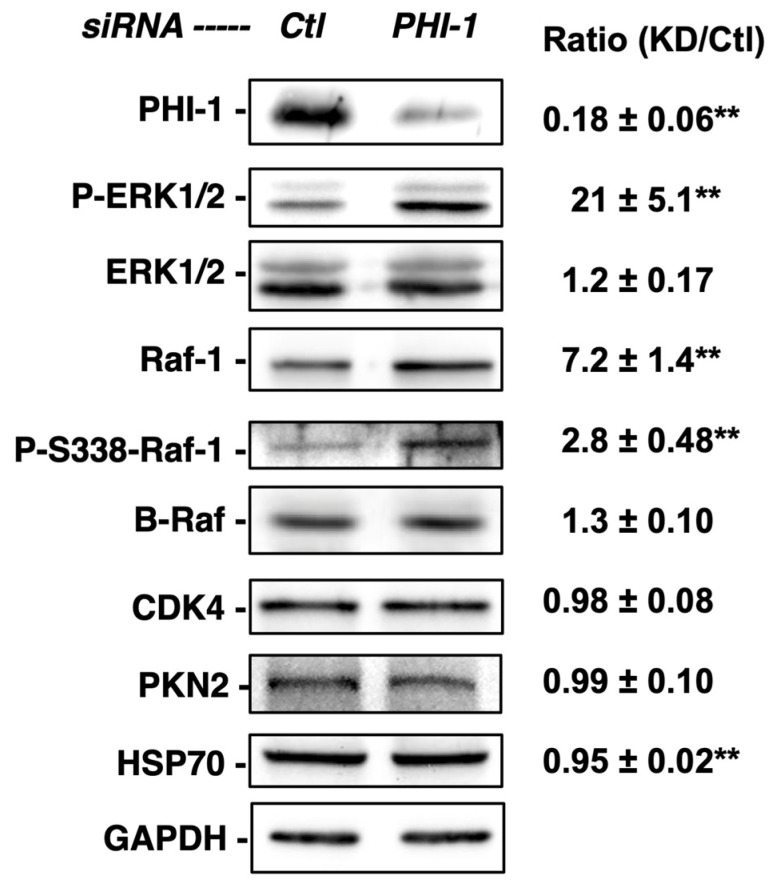
Enhanced phosphorylation of ERK1/2 in response to PHI-1 knockdown. HEK293 cells were treated with siRNA for PHI-1 or negative control (Ctl) for 72 h. Total proteins (20 µg) were subjected to immunoblotting. Density in each blot was normalized against GAPDH. Original images can be found in Appendix A. Mean values ± SEM of the relative density against the control are shown on the right. ** indicates *p* < 0.05 vs. Ctl, n = 3–7.

**Figure 3 biomolecules-13-01741-f003:**
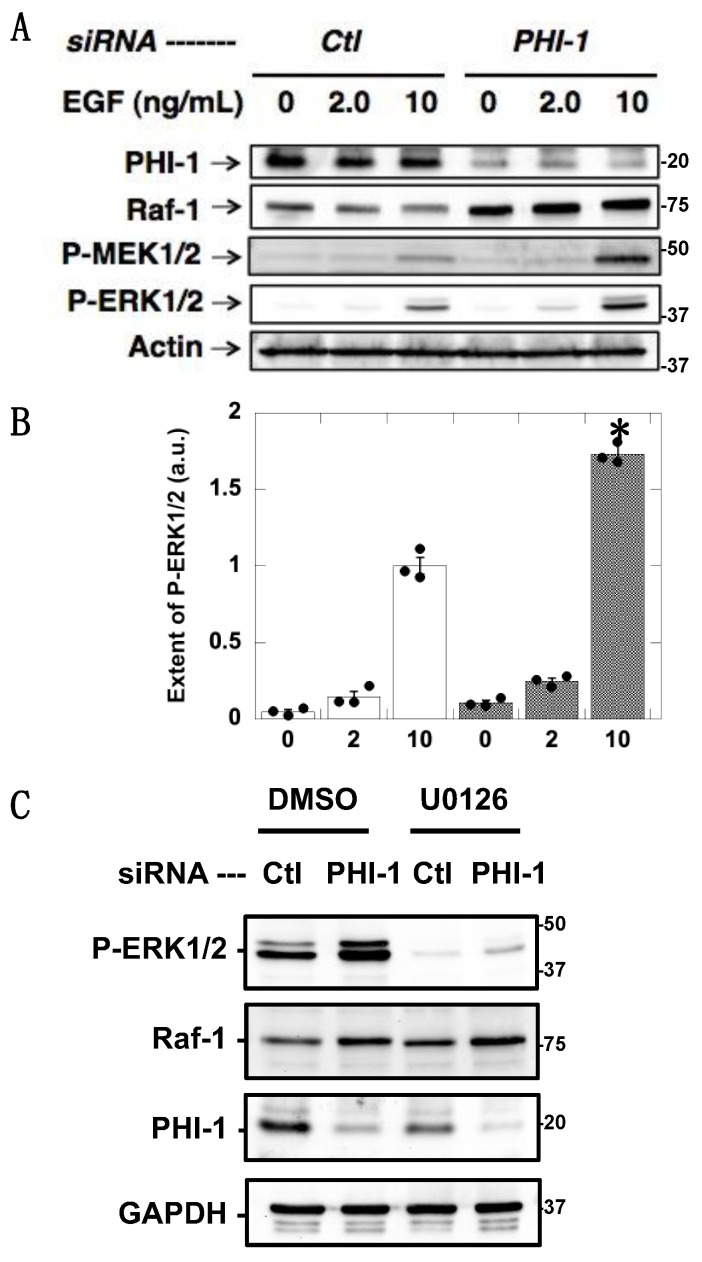
Augmentation of EGF-induced ERK1/2 phosphorylation by PHI-1 knockdown. After 24 h treatment with siRNA, HEK293 cells were harvested overnight in the serum-free medium, and then stimulated for 10 min with EGF at the indicated concentration. (**A**) Immunoblotting data. (**B**) Densitometric data of panel (**A**). * indicates *p* < 0.05, n = 3 vs. Ctl by ANOVA with Tukey’s test. (**C**) Cells were treated with 10 nM U0126 and subjected to immunoblotting. Numbers on the right indicate molecular weights in kDa. Original images can be found in Appendix A.

**Figure 4 biomolecules-13-01741-f004:**
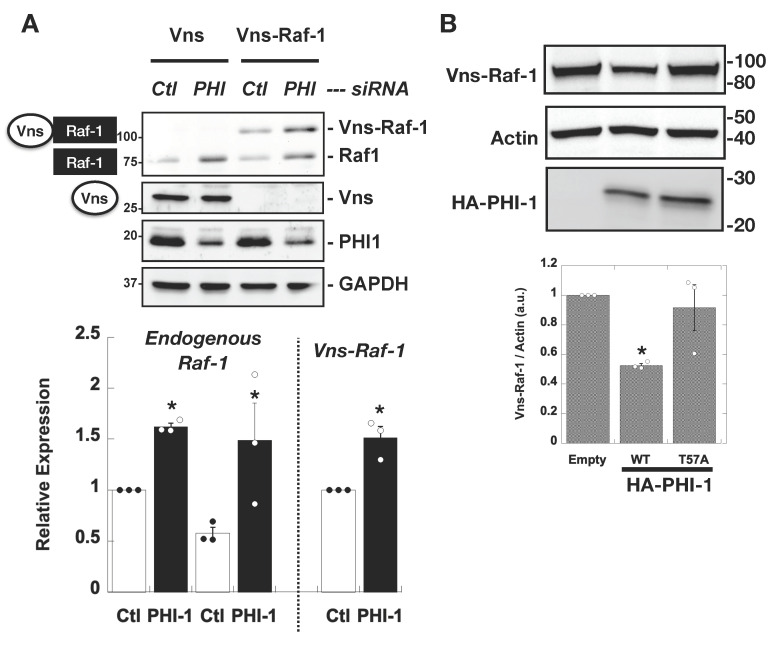
PHI-1 negatively regulates the expression of the ectopic Vns-Raf-1 fusion protein in HEK293 cells. (**A**) PHI-1 knockdown. Cells were treated for 48 h with siRNA (Ctl or PHI-1), prior to the transient transfection for 24 h. Vns or Vns-Raf-1 expression in total proteins was analyzed by immunoblotting. Numbers on the left indicate molecular weights in kDa. The bar graph indicates mean values of densitometric data from a triplicate assay. The bar graph shows mean values ± SEM of densitometric data from a triplicate assay. * indicates *p* < 0.05 vs. Ctl by *t*-test. (**B**) Co-transfection. HEK293 cells were co-transfected for 24 h with Vns-Raf-1, plus HA-PHI-1 WT, T57A, or the empty vector, and then subjected to immunoblotting. The bar graph shows mean values ± SEM of densitometric data from a triplicate assay. * indicates *p* < 0.05 vs. T57A by ANOVA with Tukey’s test. Numbers on the right indicate molecular weights in kDa. Original images can be found in Appendix A.

**Figure 5 biomolecules-13-01741-f005:**
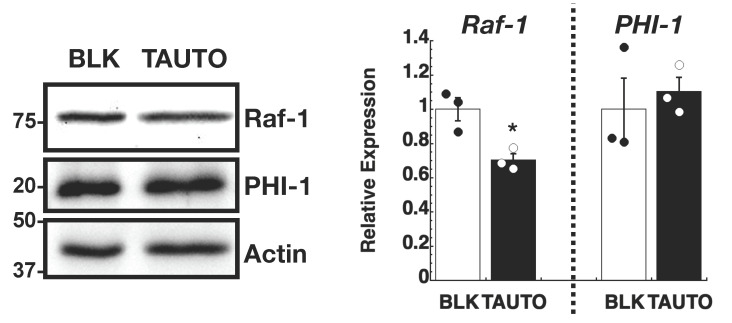
Effect of tautomycin (TAUTO) on Raf-1 expression. HEK293 cells were pre-treated for 16 h with 300 nM TAUTO or DMSO as blank (BLK) and subjected to immunoblotting (**left**) and densitometry (**right**). Numbers on the left indicate molecular weights in kDa. * indicates *p* < 0.05 vs. BLK, by *t*-test (n = 3). Original images can be found in Appendix A.

**Figure 6 biomolecules-13-01741-f006:**
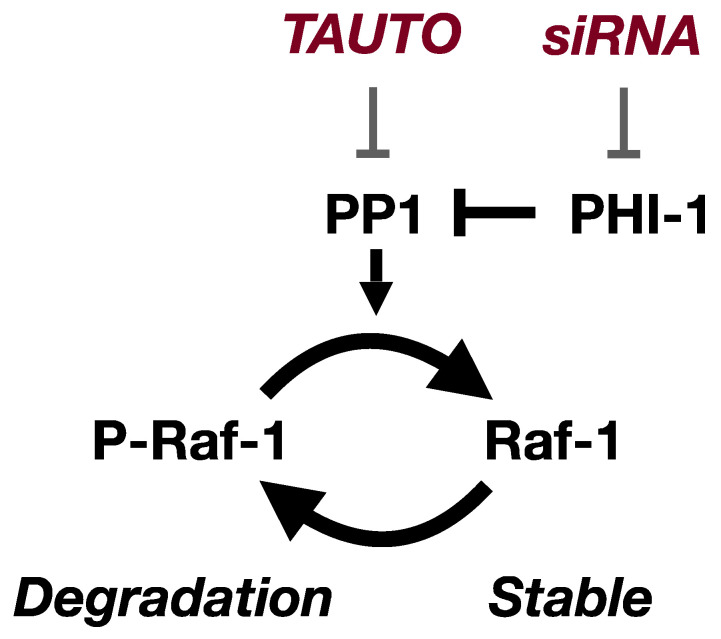
The PHI-1-PP1 pathway regulates cellular Raf-1 proteostasis in HEK293 cells. Cellular Raf-1 stability is regulated through phosphorylation at unidentified site(s). PHI-1 knockdown augments cellular PP1 activity, promoting Raf-1 dephosphorylation and subsequent stabilization. Conversely, the inhibition of PP1 by TAUTO induces a reversal of the phosphorylation state and leads to degradation.

## Data Availability

The data presented in this study are available in this article and Appendix A.

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
