# Peer review of "PHI-1, an Endogenous Inhibitor Protein for Protein Phosphatase-1 and a Pan-Cancer Marker, Regulates Raf-1 Proteostasis"

_biomolecules, 2023, doi:10.3390/biom13121741_

Round 1
Reviewer 1 Report
Comments and Suggestions for Authors
This is a well-written manuscript, supported by clearly presented and strong, convincing data, describing the interesting and novel finding that PHI-1, a widely expressed inhibitory subunit of PP1 phosphatase catalytic subunit, regulates the protein stability of Raf-1 expression.
The siRNA-induced knockdown of PHI-1leads to enhanced cell proliferation and a reduction in apoptosis within HEK293 cells. These effects are accompanied by a concurrent increase in Raf-1 expression and ERK1/2 phosphorylation. The increase in RAf-1 protein levels is attributed to elevated protein stability. These findings indicate that PHI-1 expression and activity plays a role in regulating Raf-1 activity by regulating its expression by proteostasis mechanisms. These findings under-score the influence of a novel PP1-PHI-1 signaling axis on the proteostasis of Raf-1 in cells; depending on the expression levels of PHI-1
Author Response
Thank you for the positive feedback on our manuscript. We appreciate your concise summary, highlighting the novel role of PHI-1 in regulating Raf-1 proteostasis. Your insights strengthen our work, and we are committed to further refining the manuscript based on your valuable suggestions.
Reviewer 2 Report
Comments and Suggestions for Authors
Dear Authors,
It is a very good work The experiments are neat and carried out smoothly. Its a well written manuscript, reflecting the role of the gene.
Can this be quantified "Enhanced phosphorylation of ERK1/2 in response to PHI-1 knockdown"
Need a separate conclusion model for the hypothesis in the discussion which the author is proposing instead of in fig 5.
Fig 4 A, 5A needs molecular weights labels.
Line 252 "r DMSO as black (BLK)"..it is blank or black?
What will be the predictive role of proteins as KRAS in relation to Raf1 in this set up of experiments? Was KRAS or any kinetic assay performed or tested for Raf1 activity?
Was any other cell line tested for the entire set of experiments?
Author Response
It is a very good work The experiments are neat and carried out smoothly. Its a well written manuscript, reflecting the role of the gene.
>> We appreciate the complement and all the constructive suggestions from the reviewer.
Can this be quantified "Enhanced phosphorylation of ERK1/2 in response to PHI-1 knockdown”
>>The numbers at the right side of each blot indicate the mean value ± SEM of the densitometric data from multiple blots (n=3-7). Representing blots are shown in Figure 2. To make this clearer, we rephrased a sentence in Results section at Line 182.
Need a separate conclusion model for the hypothesis in the discussion which the author is proposing instead of in fig 5.
>>As suggested, we have created a separate model illustration in Discussion section, now presented in the new Figure 6.
Fig 4 A, 5A needs molecular weights labels.
>>Corrected the figures as suggested.
Line 252 "r DMSO as black (BLK)"..it is blank or black?
>>Thank you for bringing it our attention. It should be blank and the mistake is now corrected.
What will be the predictive role of proteins as KRAS in relation to Raf1 in this set up of experiments? Was KRAS or any kinetic assay performed or tested for Raf1 activity?
>>Using the pulldown assay, we have confirmed that PHI-1 knockdown has no effect on Ras activity. The data is now shown in supplemental figure 1, and the information is included in Results section at Line 232.
Was any other cell line tested for the entire set of experiments?
>>We tested Panc1 cells, which express PHI-1 to a relatively higher extent compared to HEK293 cells (Figure 1A). PHI-1 knockdown moderately elevated Raf-1 expression as now described in Discussion section at Line349. The differential roles of PHI-1 in Panc1 cells and others are currently under investigation.